# Distributed-Framework Basin Modeling System: IV. Application in Taihu Basin

Gang Chen [1,2], Chuanhai Wang [1,2,*], Xing Fang [3], Xiaoning Li [1,2], Pingnan Zhang [2] and Wenjuan Hua [2]

1 State Key Laboratory of Hydrology-Water Resources and Hydraulic Engineering, Hohai University, Nanjing 210098, China; gangchen@hhu.edu.cn (G.C.); xzl0938@hhu.edu.cn (X.L.)
2 College of Hydrology and Water Resources, Hohai University, Nanjing 210098, China; pingnanzhang@hhu.edu.cn (P.Z.); huawenjuan0106@126.com (W.H.)
3 Department of Civil and Environmental Engineering, Auburn University, Auburn, AL 36849-5337, USA; xing.fang@auburn.edu
* Correspondence: chwang@hhu.edu.cn; Tel.: +86-136-0518-1550

**Abstract:** This paper presents the application of a distributed-framework basin modeling system (DFBMS) in Taihu Basin, China. The concepts of professional modeling systems and system integration/coupling have been summarized in the first three series papers. This study builds a hydrologic and hydrodynamic model for Taihu Basin, which is in the lowland plain areas with numerous polder areas. Digital underlying surface area data agree with the survey results from the water resource development and utilization. The runoff generated in each cell was calculated with the model based on the digital underlying surface data. According to the hydrological feature units (HFU) concept from the DFBMS, Taihu Basin was conceptualized into six different HFUs. The basic data of rainfall, evaporation, water surface elevation (WSE), discharge, tide level, and water resources for consumption and discharge in 2000 were used to calibrate the model. The simulated results of WSE and discharge matched the observed data well. The observed data of 1998, 1999, 2002, and 2003 were used to validate the model, with good agreement with the simulation results. Finally, the basic data from 2003 were used to simulate and evaluate the management scheme of water diversion from the Yangtze River to Taihu Lake. Overall, the DFBMS application in Taihu Basin showed good performance and proved that the proposed structure for DFBMS was effective and reliable.

**Keywords:** Taihu Basin; distributed-framework basin model application; hydrological feature units; polder areas





## 1. Introduction

The hydrological cycle in a basin is a complex process that is influenced by meteorological conditions, physical characteristics of the underlying surface, and human activities [1–5]. Urbanization in China has been in a state of acceleration since 1978 [6]. Urbanization of river systems has been widely recognized as the most significant influence of human activities, leading to hydrological regime variations and uncertainties in flood control [7]. With impervious areas associated with buildings and increasing transportation infrastructure [8,9], both surface runoff and flow velocity into rivers have increased significantly, leading to frequent regional flooding [10]. In order to meet the demand of land for urbanization [11], rivers have also been buried and diverted to a certain extent, resulting in a great change in river system structures. This has had an impact on water transfer, storage, and the adjustable capacity of the basin [12–14].

Taihu Basin is the core economic region in the Yangtze River Delta of China. Flood control and water pollution are significant problems in Taihu Basin due to the dramatic urbanization over the last 20 years [15,16]. The environment and underlying surface conditions have been affected to a large extent by human activities [17]. Meanwhile, water

demands for industry, urban living, and agriculture have increased year by year. Groundwater overdraft leads to continuous land-surface subsidence in the plains areas of Taihu Basin [15]. According to a survey, land subsidence for the economic core of Suzhou, Wuxi, and Changzhou in Jiangsu province has reached 1 m [18]. The funnel-shaped groundwater surface area was 7000 km$^2$. On the one hand, Taihu Basin is low in water resources due to a large amount of withdrawal in the dry year and water quality deterioration in the rivers and lakes [19,20]. On the other hand, frequent flooding in urban areas often happens in rainy years. Water resources and the environment have become the main factors restricting socioeconomic development in Taihu Basin. In particular, blue-green algal blooming in Taihu Lake has caused a crisis of urban water supply around the lake [21,22]. This reflects water environment problems in the whole of Taihu Basin.

Following a basin-wide catastrophic flood in 1991, the first round of comprehensive regulation projects began in China [23]. The water resources and environment in Taihu Basin are basin-wide problems that influence the river network and lakes in the whole basin. Human activities including water diversion and drainage along the Yangtze River, the inflow and outflow of Taihu Lake, and urban flood control projects need to be considered to identify the appropriate hydrological response at a basin scale. The water cycles and dynamic water conditions in the Taihu Basin are the research basis for water environment problems [24]. Strategies and measures for improving the water environment can be proposed through the study of the mechanisms by which pollutants have occurred, transported, and decayed. Hence, basic research is urgently needed on the Taihu Basin digital watershed model to study the water cycle, water environment, and water ecology.

This paper is the last in a series of papers on the distributed-framework basin modeling system (DFBMS), which is the application of DFBMS in the Taihu Basin. We built a hydrologic and hydrodynamic model for the whole Taihu Basin, which is a lowland plain area with lots of polder areas. The study area is divided into hilly sub-watersheds, hilly rivers, plain overland flows, plain rivers, lakes and reservoirs, and hydraulic engineering structures based on the hydrological feature units (HFU) concept. The calibration and validation results proved that the DFBMS accurately reflects runoff generation and water movement for Taihu Basin. Through this series of studies, we hope to direct the sustainable use of water resources and socio-economic development for the whole basin. Our results can also improve response and treatment for real-time flood forecasting and management.

## 2. Materials and Methods

### 2.1. Study Area

Taihu Basin is in the south of Changjiang Estuary and north of Qiantang River, China (Figure 1a). It is the last sub-basin on the right bank of Changjiang River. Taihu Basin covers 36,895 km$^2$ area and includes parts of Jiangsu, Zhejiang, Anhui Province, and Shanghai. The terrain of Taihu Basin is high in the west and low in the east. The western part is a hilly area of 7338 km$^2$, and the eastern part is a plain area of 29,557 km$^2$. The plain area can be further divided into the middle plain area with an altitude elevation less than 5 m and the eastern plain area with an elevation from 5 to 12 m. That is to say, the terrain of Taihu Basin is like a bowl with higher elevations around and lower elevations in the middle. These natural conditions have caused difficulty in water drainage and many flood disasters.

Taihu Basin is a typical river network area with many lakes, dams, control gates, pump stations, and culverts. The flow moves alternately backward and forwards, which is not only controlled by the Changjiang River and Qiantang River but also by the hydraulic structures in the basin. There are 189 lakes with a water surface area larger than 0.5 km$^2$ and 10 large lakes of more than 10 km$^2$. The lakes are all shallow with an average depth of less than 2 m. The total surface area and water for all lakes in Taihu Basin are 3159 km$^2$ and 5.78 billion m$^3$, respectively.

In Taihu Basin, there are lots of polder areas (the green in Figure 1b) that cover more than 8541.4 km$^2$. A polder area is a low-lying and stagnant field in the plain river network. When the floodwater level is higher than ground elevation, the embankment around the

polder areas is needed to protect these areas from outside flooding. Internal waterlogging can be drained using a water gate and pump station. In this way, the rivers outside and inside of these polders are connected according to the operation rule, which is usually based on the outside and inside water level. As required, the gate can be open for normal production and living. Otherwise, the gate will be closed, and the water is drained through the pump.

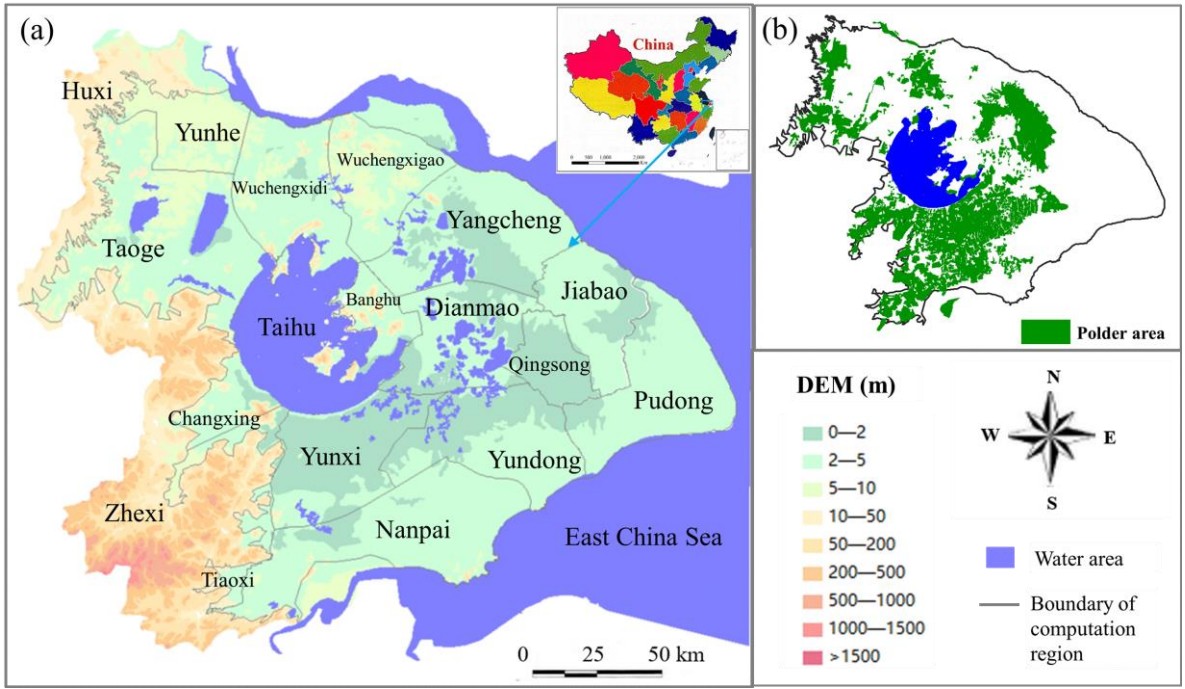

**Figure 1.** (**a**) Location of Taihu Basin in China and its digital elevation model (DEM); (**b**) the polder area in Taihu Basin.

### 2.2. Statistics of Underlying Surface Information

According to the basin terrain, river network, characteristics of water resources, and comprehensive planning of watershed management, the study area can be further subdivided into 18 computational regions, including 16 plain areas and two hilly regions (Figure 1a).

The underlying surface of the plain area can be classified as water, paddy field, rain-fed land, and construction land according to the Integrated Water Resources Plan of the plain area in Taihu Basin. The scale of the digital map for the construction land, water, and administrative divisions is 1:250,000. Based on statistical analysis, the areas of water, paddy field, rain-fed land, and construction land outside and inside the polder area in 16 computational regions are shown in Table 1.

**Table 1.** The areas of different underlying surfaces of the plain area in Taihu Basin, including outside and inside the polder area.

| | Subarea | Water Area (km²) | Paddy Field (km²) | Rain-Fed Land (km²) | Construction (km²) | Total (km²) | Lakes (km²) | Total (km²) |
|---|---|---|---|---|---|---|---|---|
| 1 | Yunhe | 25.59 (13.35) [1] | 468.36 (93.45) | 161.31 (73.43) | 259.21 (17.8) | 914.47 (198.03) | 0 | 1112.5 |
| 2 | Taoge | 104.66 (105.74) | 938.05 (648.11) | 607.18 (405.92) | 450.27 (115.98) | 2100.16 (1275.75) | 294.4 | 3670.3 |
| 3 | Wuchengxidi | 56.09 (52.85) | 196.3 (481.31) | 441.67 (154.78) | 462.44 (20.76) | 1156.5 (709.70) | 21.3 | 1887.5 |
| 4 | Wuchengxigao | 85.17 (3.03) | 747.52 (25.72) | 267.84 (4.54) | 358.63 (6.05) | 1459.16 (39.34) | 20.8 | 1519.3 |
| 5 | Yangcheng | 92.93 (132.48) | 321.4 (726.21) | 250.25 (377.82) | 483.32 (68.7) | 1147.9 (1305.21) | 166.2 | 2619.3 |
| 6 | Dianmao | 120.65 (78.14) | 304.75 (543.85) | 43.76 (15.63) | 373.51 (62.51) | 842.67 (700.13) | 230.9 | 1773.7 |

**Table 1.** *Cont.*

| | Subarea | Water Area (km²) | Paddy Field (km²) | Rain-Fed Land (km²) | Construction (km²) | Total (km²) | Lakes (km²) | Total (km²) |
|---|---|---|---|---|---|---|---|---|
| 7 | Banghu | 33.66 (6.88) | 5.36 (45.14) | 476.6 (74.21) | 106.33 (16.83) | 621.95 (143.06) | 0 | 765 |
| 8 | Yunxi | 118.28 (120.35) | 110.64 (692.97) | 106.76 (394.04) | 234.87 (133.94) | 570.55 (1341.3) | 54.6 | 1968.8 |
| 9 | Yundong | 63.49 (154.54) | 20.6 (904.56) | 28.85 (498.64) | 131.87 (195.75) | 244.81 (1753.49) | 78.6 | 2076.9 |
| 10 | Nnanpai | 96.39 (134.75) | 246.07 (872.96) | 210.92 (729.42) | 445.27 (161.12) | 998.65 (1898.25) | 32.5 | 2929.4 |
| 11 | Jiabao | 159 (0) | 399.62 (0) | 97.09 (0) | 751.39 (0) | 1407.1 (0) | 0 | 1407.1 |
| 12 | Qingsong | 69.76 (0) | 441.33 (0) | 65.97 (0) | 181.23 (0) | 758.29 (0) | 0 | 758.3 |
| 13 | Pudong | 225.53 (0) | 856.08 (0) | 547.71 (0) | 671.98 (0) | 2301.3 (0) | 0 | 2301.3 |
| 14 | Taihu | 0.09 (0) | 43.79 (0) | 9.26 (0) | 35.87 (0) | 89.01 (0) | 2338 | 2427 |
| 15 | Changxing | 26.47 (28.99) | 23.95 (77.51) | 87.6 (256.49) | 78.14 (51.05) | 216.16 (414.04) | 0 | 630.2 |
| 16 | Tiaoxi | 48.31 (22.08) | 23.46 (37.27) | 217.38 (238.08) | 47.62 (55.9) | 336.77 (353.33) | 0 | 690.1 |
| | Total | 2179 | 10,296 | 6843 | 5978 | 25,299 | 3237 | 28,536 |

Note: [1] The first number is the area outside of the polder area; the number in parentheses is the area inside of the polder area.

### 2.3. Study Area Digitization

Taihu Basin was divided into 274 × 237 computation cells with a mesh size equal to 1 km × 1 km. The attributes of each cell can be realized by overlaying them on the 16 computational regions in the plain area. The statistical areas of the 16 computational regions are listed in Table 2. Due to the large range of computational regions in the basin, the model set for each grid belongs to the same region. There are some cells covering two zones that can be treated as one region according to the weight coefficient of the area. Overall, there was high consistency between the statistics and digitization results, based on the small relative error for each computational region (Table 2).

**Table 2.** Summary of area for 16 computational regions in the plain area.

| | Subarea | Mesh Area (km²) | Survey Results from Integrated Water Resources Plan (km²) | Error (%) |
|---|---|---|---|---|
| 1 | Yunhe | 1109 | 1112.5 | −0.31 |
| 2 | Taoge | 3666 | 3670.3 | −0.12 |
| 3 | Wuchengxidi | 1886 | 1887.5 | −0.08 |
| 4 | Wuchengxigao | 1523 | 1519.3 | 0.24 |
| 5 | Yangcheng | 2596 | 2619.3 | −0.89 |
| 6 | Dianmao | 1765 | 1773.7 | −0.49 |
| 7 | Banghu | 734 | 765.0 | −4.05 |
| 8 | Yunxi | 2000 | 1968.6 | 1.60 |
| 9 | Yundong | 2102 | 2076.9 | 1.21 |
| 10 | Nnanpai | 2958 | 2929.4 | 0.98 |
| 11 | Jiabao | 1410 | 1407.1 | 0.21 |
| 12 | Qingsong | 768 | 758.3 | 1.28 |
| 13 | Pudong | 2313 | 2301.3 | 0.51 |
| 14 | Taihu | 2459 | 2427.0 | 1.32 |
| 15 | Changxing | 629 | 630.2 | −0.19 |
| 16 | Tiaoxi | 687 | 690.1 | −0.45 |
| Total | plain area | 28,605 | 28,536.5 | 0.24 |

According to the concept of the hydrological feature unit (HFU) for plain overland flow in the second paper in the series, the underlying surface of Taihu Basin was digitized as water, paddy field, rain-fed land, and construction land for the DFBMS (Figure 2), which was the same classification as in the Integrated Water Resources Plan. The western part of Taihu Basin consists of hilly areas that were mostly classed as woodland.

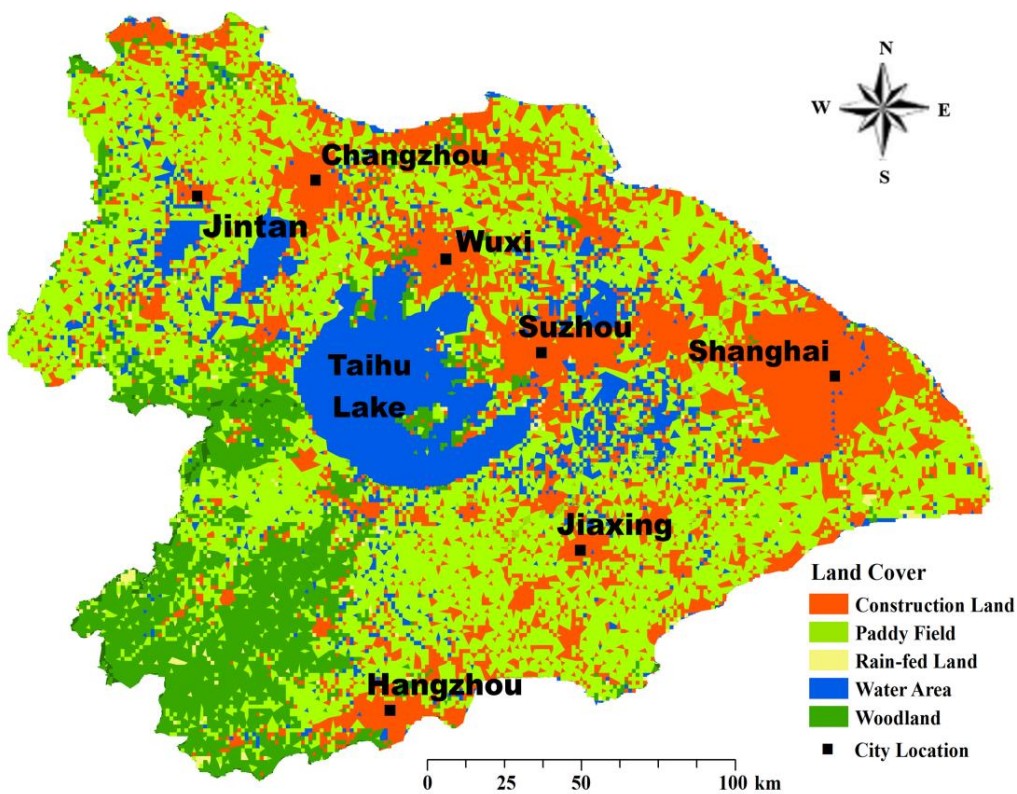

**Figure 2.** The digital land cover for the distributed-framework basin modeling system in Taihu Basin.

In the DFBMS, the water body can be simulated as a storage node, generalized channel, or lake. A larger water body (such as a large lake) can be easily extracted from the satellite picture and electronic map. However, it is hard to distinguish a small water body in the basin. The area of the generalized channel in the digital model is the size of the cross section. The channel area is estimated when the water surface elevation is set to 3 m. There are also 76 storage nodes in the simulation model of Taihu Basin. The total water area for the mesh is smaller than the actual water area. Four kinds of digital underlying surface areas are shown in Table 3, which agree with the survey results in Table 1. After we obtained the underlying surface information for each grid, the runoff on each surface was calculated based on the rainfall-runoff mechanism introduced in the second paper in the series.

### 2.4. Distributed-Frame Hydrological and Hydraulic Modeling for Taihu Basin

#### 2.4.1. Model Structure

The Taihu Basin model is multibasin, multiscale, and multiprocess, covering not only hydrological but also hydrodynamic processes in the hilly and plain areas. The application of the distributed-frame hydrological and hydraulic models in Taihu Basin includes different corresponding HFUs. Figure 3 shows the logical architecture diagram of the distributed-frame professional modeling system in the developed Taihu Basin model. In the upstream part, there are the Zhexi Mountains and Huxi hilly regions. The runoff generation and confluence were simulated based on the rainfall and underlying surface data (The mechanisms are described in the second and third series papers). The hydrograph routing from the upstream arrived at the river network in the plain area and then was discharged to a zero-dimensional lake, two-dimensional lake, one-dimensional river, or hydraulic engineering structure. The tidal elevations along the Yangtze River and Hangzhou Bay were set as one kind of downstream boundary. The hydraulic engineering operation rule was also used as another boundary condition.

**Table 3.** The digital area of four kinds of underlying surface (outside and inside of polder area).

| Subarea | Water Area | Paddy Field | Rain-Fed Land | Construction | Total |
|---|---|---|---|---|---|
| Yunhe | 11.95 (1.18) [1] | 468.36 (93.45) | 161.31 (73.43) | 202.84 (8.45) | 844.46 (176.51) |
| Taoge | 27.98 (23.07) | 938.05 (648.11) | 607.18 (405.92) | 367.29 (100.14) | 1940.50 (1177.24) |
| Wuchengxidi | 30.52 (8.00) | 196.3 (481.31) | 441.67 (154.78) | 116.63 (0.00) | 785.12 (644.09) |
| Wuchengxigao | 64.40 (0.06) | 747.52 (25.72) | 267.84 (4.54) | 245.03 (3.65) | 1324.79 (33.97) |
| Yangcheng | 62.54 (20.88) | 321.40 (726.21) | 250.25 (377.82) | 385.92 (0.00) | 1020.11 (1124.91) |
| Dianmao | 39.57 (32.17) | 304.75 (543.85) | 43.76 (15.63) | 276.48 (33.35) | 664.56 (625) |
| Banghu | 8.91 (1.84) | 5.36 (45.14) | 476.60 (74.21) | 45.33 (4.75) | 536.20 (125.94) |
| Yunxi | 54.75 (45.87) | 110.64 (692.97) | 106.76 (394.04) | 220.63 (110.88) | 492.78 (1243.76) |
| Yundong | 12.78 (24.05) | 20.60 (904.56) | 28.85 (498.64) | 126.97 (160.19) | 189.20 (1587.44) |
| Nnanpai | 55.74 (33.88) | 246.07 (872.96) | 210.92 (729.42) | 387.26 (80.05) | 899.99 (1716.31) |
| Jiabao | 103.01 (0.00) | 399.62 (0.00) | 97.09 (0.00) | 232.99 (0.00) | 832.71 (0.00) |
| Qingsong | 45.70 (0.05) | 441.33 (0.00) | 65.97 (0.00) | 143.21 (0.00) | 696.21 (0.05) |
| Pudong | 165.82 (0.00) | 856.08 (0.00) | 547.71 (0.00) | 521.02 (0.00) | 2090.63 (0.00) |
| Taihu | 0.00 (0.00) | 43.79 (0.00) | 9.26 (0.00) | 34.82 (0.00) | 87.87 (0.00) |
| Changxing | 15.59 (2.80) | 23.95 (77.51) | 87.60 (256.49) | 73.71 (46.08) | 200.85 (382.88) |
| Tiaoxi | 31.82 (7.75) | 23.46 (37.27) | 217.38 (238.08) | 36.73 (48.08) | 309.39 (331.18) |

Note: [1] The first number is the area outside of the polder area; the number in parentheses is the area inside of the polder area.

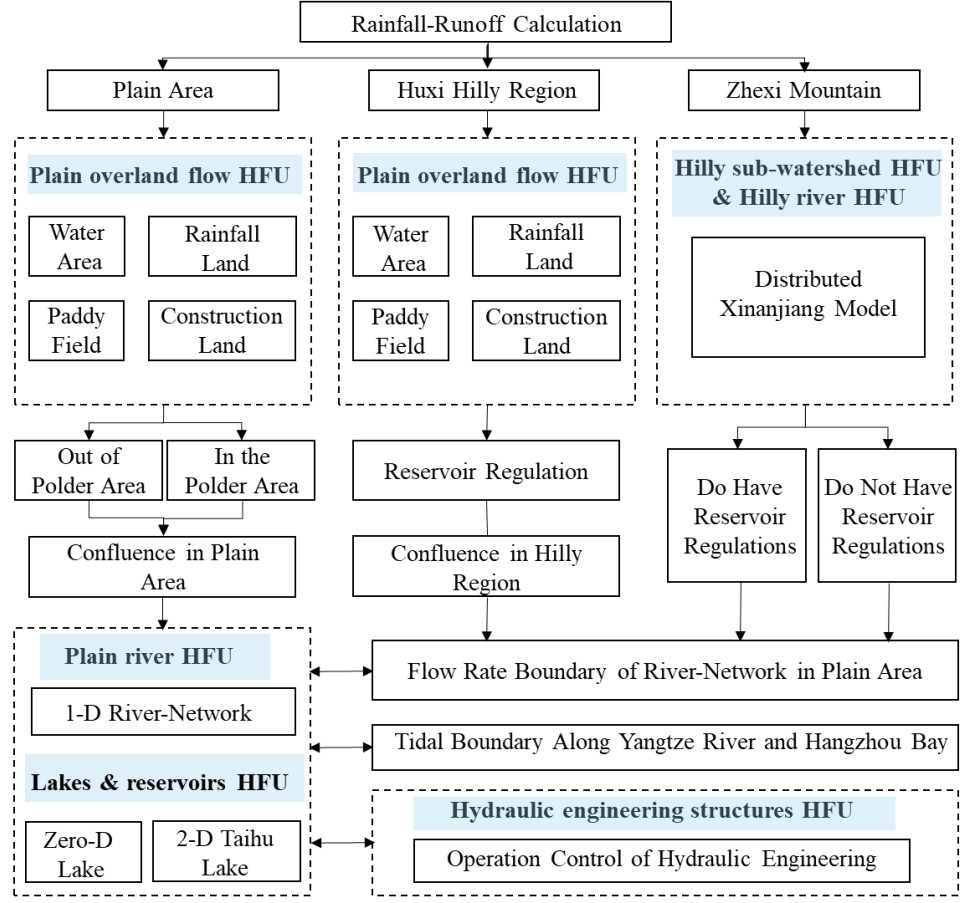

**Figure 3.** The logical architecture diagram of the distributed-frame hydrological model in Taihu Basin.

2.4.2. Hydrological Feature Units in Taihu Basin

Taihu Basin is a typical plain tidal river area with hilly areas in the upstream part and a flat plain in the middle part (Figure 1a). Hence, the distributed-framework basin modeling system (DFBMS) proposed in this series of papers was applied to solve the complex rainfall-runoff generation and flow movement process in the basin. According to the concept of

HFU from the distributed-frame professional modeling system (DF-PMS), Taihu Basin can be categorized into six feature units: (1) hilly subwatershed, (2) hilly river, (3) plain overland flow, (4) plain river, (5) lakes and reservoirs, and (6) hydraulic engineering structures. Table 4 shows the summary of the main HFUs in the Taihu Basin model.

**Table 4.** Main generalized information for hydrological feature units (HFUs) in Taihu Basin.

| Type of Hydrological Feature Unit (HFUs) | Model Algorithm | Generalization Details |
|---|---|---|
| Hilly subwatershed and Hilly river | Xinanjiang model from 2nd paper | Hilly area of Zhexi, divided into 10 subcatchments; Hilly area of Huxi, divided into 10 subcatchments; |
| Plain overland flow | Rainfall-runoff model in delta plain from 2nd paper | Mainly for delta plain, divided into 16 subdomains |
| Plain river | One dimensional hydrodynamic model from 3rd paper | 1482 rivers of 7879.4 km; the number of cross sections is 4270 |
| Lakes and reservoirs (including flood plains and paddy fields) | Zero-dimensional model from the 3rd series paper Two-dimensional model for Taihu Lake area from the 3rd paper | 104 lakes of 871 km$^2$ Taihu Lake area |
| Hydraulic engineering structures | Gate and dam simulation from 3rd paper | 168 gates |

Figure 4a is the distributed map of hilly subwatershed HFUs in the hilly area. Zhexi and Huxi hilly areas are both divided into 10 subcatchments. The plain area is set as 16 plain overland flow HFUs according to the computational region (Figure 4b). The runoff for meshed underlying was calculated according to four digital surfaces in the DFBMS. Regardless of whether it is inside or outside of the polder area, the same kind of underlying surface has the same runoff mechanism. Finally, the overland flow discharges to the river network. Figure 4c shows the generalization of the river network, lakes, and gates (in green). The DFBMS generalizes 1482 one-dimensional rivers of 7879.4 km. It also has 104 zero-dimensional lakes and one quasi-three-dimensional lake (i.e., Taihu lake), and 168 main controlling gates simulated in the DFBMS. Figure 4d shows the river network polygon, which consists of a generalized river network for Taihu Basin. That is, those closed subareas are formed by the surrounding river reach. The runoff process of each kind of underlying surface flow into a certain river reach can be computed through the concept of the minimum flow path. The detailed information is from the second paper in the series. In this way, hydrologic and hydrodynamic models were easily coupled.

2.4.3. Polder Areas Simulation

Polder areas make up a large proportion of Taihu Basin (Figure 1b). The simulation between the outside and inside polder areas shows the special cases in this study area that are described in Figure 5. $A_0$ and $A_I$ are storage areas for outside and inside polder areas, respectively. $Z_0$ and $Z_I$ are the water surface elevation of the storage area for outside and inside polder areas, respectively. $O(t)$ and $I(t)$ are the runoff process for outside and inside polder areas, respectively. In this study, the concept of a lateral inflow rate is proposed to calculate the virtual connection between the outside and inside polder areas. That is, the water exchange between the river network and polder areas was through a small ditch or stream. The ratio of total stream width to length of the passing river is set as a parameter equal to 0.1 in the model. The bottom elevation of the corresponding reach was used as the height of the virtual connection bottom. The storage unit between the outside and inside polder areas built the hydraulic connection through the virtual connection. For the outside of the polder area, the virtual connection is always open in the whole simulation. For the inside of the polder area, the virtual connection is open or closed based on the different

situations. The simulation of virtual connection produces the broad crested weir shown in the hydraulic modeling in the third paper in the series. The distributions of generalized polder areas in Taihu Basin are shown in Figure 1b.

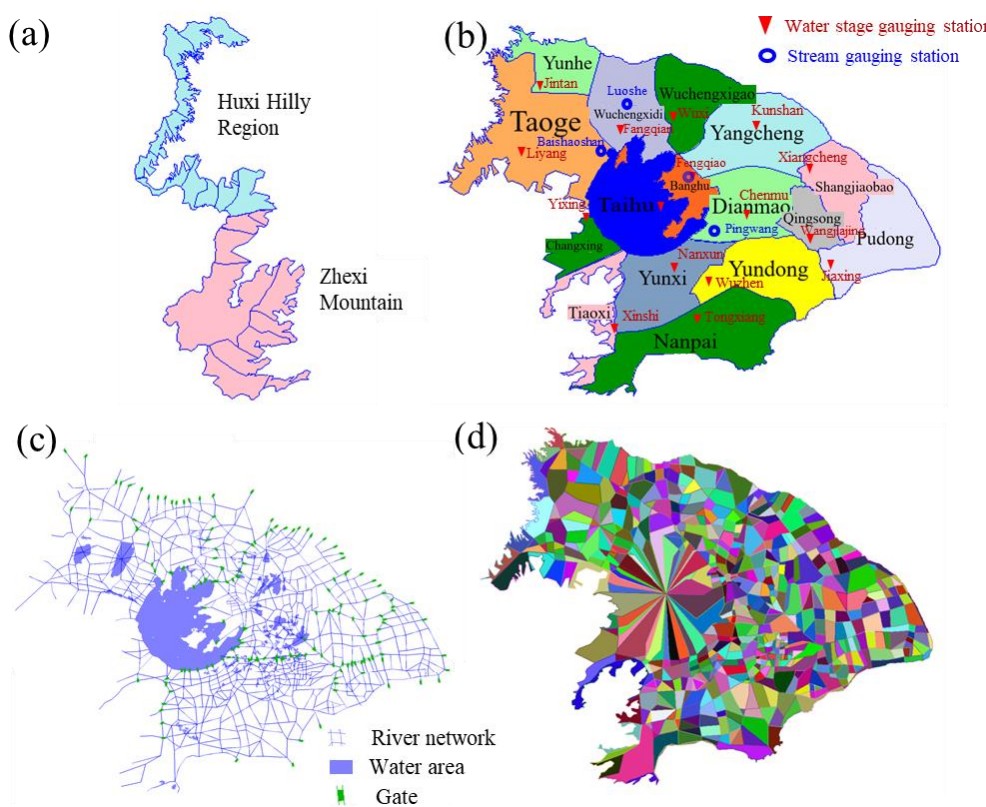

**Figure 4.** The distribution map of overland runoff units in (**a**) hilly and (**b**) plain areas including water stage gauging station and stream gauging station; (**c**) the generalization of the river network, lakes, and gates (in green) in Taihu Basin, and (**d**) the river network polygon for Taihu Basin.

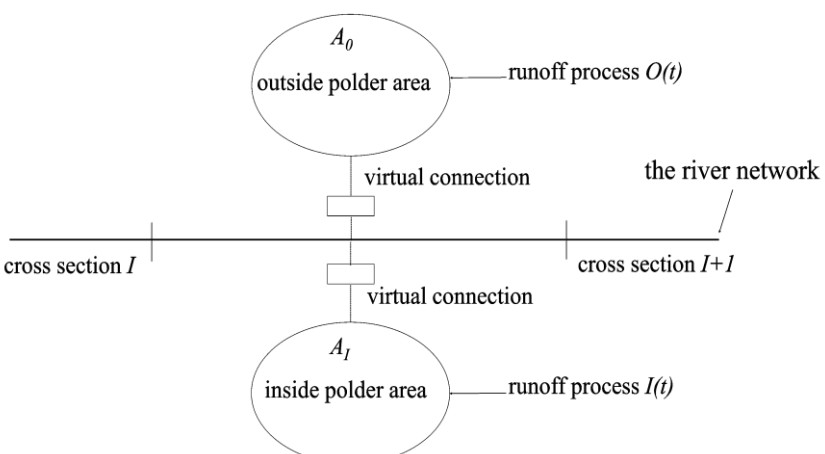

**Figure 5.** The regulation relationship between outside and inside polder area.

The operation rule is as follows:

1. When the water surface elevation outside the polder area is higher than the critical value, the virtual connection is closed. When the water depth of the storage area inside the polder area is higher than 0.4 m, the drainage pump is switched on.

2.　When irrigation in the polder area is needed and there is no rain, the water inside the polder area is used until the depth reduces to 0 m, then the water outside the polder area can be used through the pump.

### 2.4.4. Rainfall, Water Supply, and Drainage Simulation

In the study area, 114 rainfall stations and 12 evaporation stations were used to generate a Thiessen polygon to calculate the average rainfall and evaporation for different computational regions (Figure 6). Thiessen polygons are used to allocate space rainfall to the nearest point feature. The calculated precipitations for different underlying surfaces of the whole basin are the input data for the runoff generation. The adjacent evaporation station was adopted when there was no evaporation in the computation region.

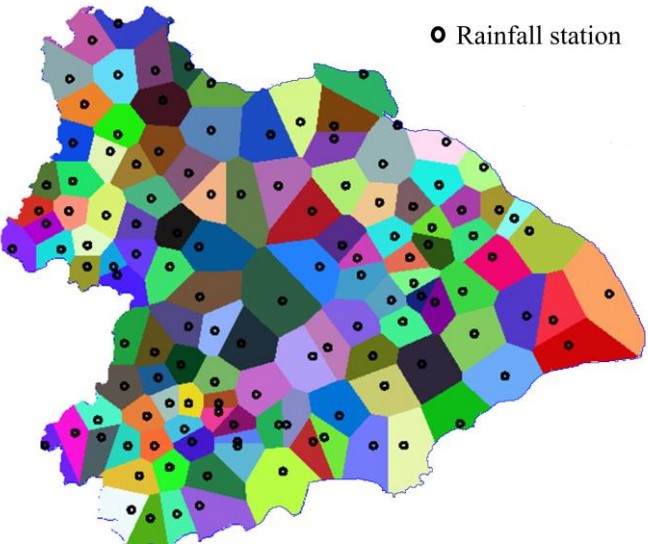

**Figure 6.** Distribution of precipitation stations, and Thiessen polygon generated by DFBMS.

The Integrated Water Resources Plan for the Taihu Basin model in 2000 provided a large amount of specific basic data of water supply, usage, and drainage. The water supply and drainage systems were generalized in the model and are directly influential on the water flow in the river network. There was no need to deal specifically with water usage after we considered the water supply and drainage system.

For the water supply of industry and urban living, the specific geographic location of the water supply company and self-contained water source can be set as the water intake. The water quantity will be assigned to the nearest river. If there is no specific geographic location for the water supply company or self-contained water source, the water quantity was taken to represent the average processing in a certain area. Taking the prefecture-level region as a control unit, the difference in quantity between water supply and water intake can also be treated as the average intake. According to a survey report in 2000, there is 3.78 billion $m^3$ water from 133 water supply companies and 11.28 billion $m^3$ water from 102 self-contained water sources (Figure 7a).

Agricultural supply includes irrigation of paddy fields, drylands, and drinking water in the countryside. The irrigation water of the paddy fields is already considered in the hydrologic model of DFBMS. The water from the river network is taken for the paddy field. The water consumption for this part was 6.82 billion $m^3$ in 2000. The irrigation water from the survey report was used for dryland to calculate the area-average intake for the corresponding region. The same as dryland, the consumption for the living water of the countryside was simulated using the area-average method. The water consumption for irrigation water and drinking water in the countryside was 0.91 billion $m^3$ in the model.

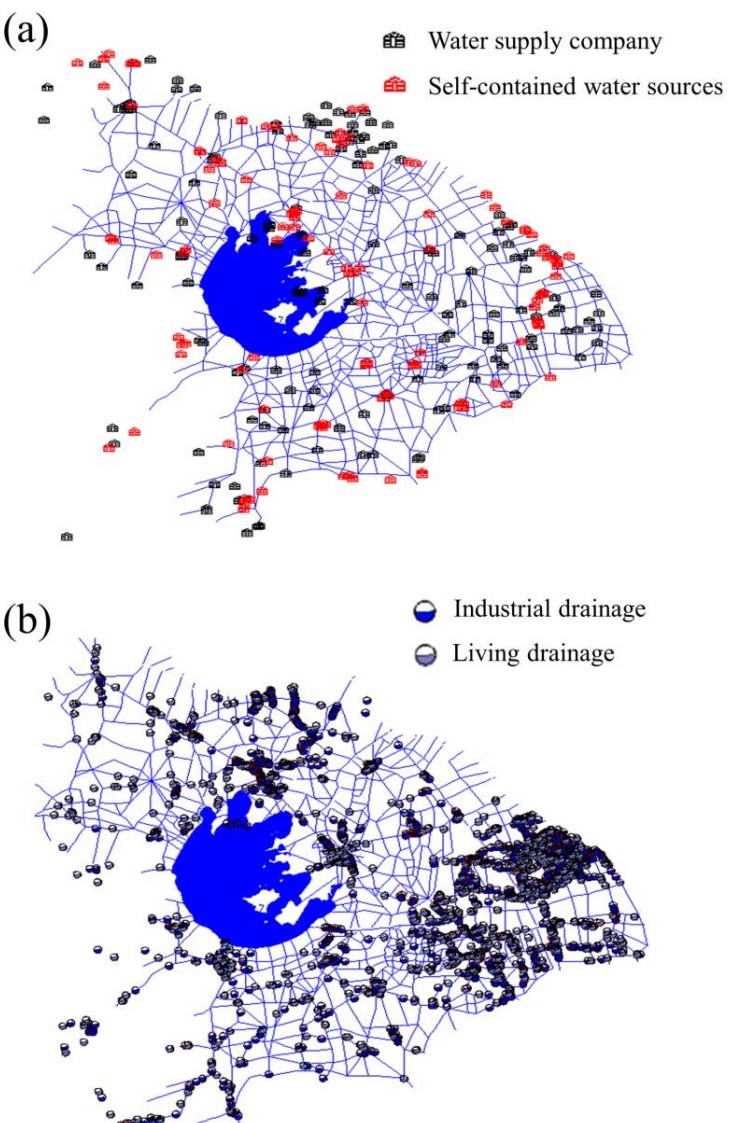

**Figure 7.** The simulations of distribution (**a**) for water supply companies (black) and self-contained water sources (red); (**b**) drainage in Taihu Basin.

The drainage for industry and drinking water in towns is the same as the water supply simulation. That is to say, drainage with a specific geographic location can be set as a drain point; otherwise, it can be dealt with like the area-average results. According to the report from 2000, the model simulated about 2000 sewage discharge points with 3.97 billion m$^3$ in the study area (Figure 7b).

## 3. Results and Discussion

### 3.1. Model Calibration

The data for 2000 were sufficient for model calibration, which includes rainfall, evaporation, water surface elevation, discharge, tide level, and water resource consumption and discharge. In this case, the data from 2000 were used to calibrate the developed model in Taihu Basin. DFBMS took about 7 min to simulate the whole Taihu basin for one year.

3.1.1. Runoff Calculation for Taihu Basin

The runoff calculation process of Taihu Basin in DFBMS is divided into two parts. One is for the plain area with 16 computational regions, calculated based on the plain overland flow HFU principle. The other part is hilly areas with 20 computational regions calculated

based on the hilly subwatershed HFU. In the study area, 114 rainfall stations were used to generate a Thiessen polygon to calculate the average rainfall for different computational regions (Figure 6). The 12 evaporation stations were applied in the whole basin. The total runoff of Taihu Basin calculated with the DFBMS was 7.59 billion m$^3$ for 2000.

The simulated total runoff of Taihu Basin is calibrated by the calculation of basin water balance. The objective was to calculate the change of waterbody storage for the plain area, which is shown in Equation (1):

$$DV = R + SR - SH - W + WD - WS \tag{1}$$

where *DV* is the change of the amount of waterbody storage in rivers and lakes for the plain area. For each computational region, representative water stage gauging stations are chosen to calculate the daily mean water surface elevation, which is called the representative mean water level. Based on the representative mean water level, the water storage for distributed waterbodies in the computation region can be calculated and totaled based on the stage-volume curve. Then the variation of daily storage for the whole basin is obtained, where positive and negative means a storage increase and decrease, respectively; *R* is the daily runoff to calibrate, including Huxi, Zhexi hilly area, and the plain area; *SR* is the measured discharge for diversion and drainage along the Yangtze River and Hangzhou Bay; and *WD* and *WS* are the amount of wastewater and withdrawal in the plain area, respectively. These two data points were taken from the Integrated Water Resources Plan of Taihu Basin. Hence, *R*, calculated based on the basin water balance Equation (1), was 7.23 billion m$^3$ in 2000. Compared to the modeling results from DFBMS, the percentage difference between the simulation and calculation was 4.7%. The runoff generations are calculated on water area, rain-fed land, paddy field, and construction land. For the rain-fed and construction land, there is no runoff when there is no rain. But for the paddy field, the runoff will be negative under irrigation demand. The paddy fields need water from May to September, which gives the most negative values in Figure 8. The runoff will also be negative for the water area when there is only evaporation.

In the runoff simulation of paddy fields and rain-fed land in Taihu Basin, infiltration losses are considered in different ways. Take paddy fields as an example, the parameters are established based on the analysis of the agricultural experimental site in Taihu Basin. There are 205 hm$^2$ seedling fields in the whole survey area. The seedling period in the test area was 41 days from sowing on 16 May to transplanting on 25 June. In the seedling stage, the gross irrigation amount is 782 m$^3$/hm$^2$, the net irrigation amount is 368 m$^3$/hm$^2$. The water loss of canals is equivalent to 414 m$^3$/hm$^2$, the average daily water loss can be calculated as 414/666.7/41 = 15.1 mm. The daily infiltration of seedling fields is 7.7 mm, then the total infiltration loss is 22.8 mm per day. For rain-fed land, the infiltration losses are processed in a new way. This infiltration water is used to wet and increase the soil water content of the surrounding dry land. With the increase of soil water content in the dry land, the evaporation of soil water increases when there is no rain. And the runoff generation of dry land will increase when there is rain. It is worth mentioning that there are no observed data for the runoff process in 16 computational regions. The parameters of the hydrologic model for subwatershed and plain overland flow HFU are calibrated according to the water surface elevation or discharge data from representative stations in different regions. The runoff of time series and cumulative curve followed the same trend (Figure 8). In terms of conservation of mass, this proves that the DFBMS can simulate rainfall-runoff in complex geographical conditions.

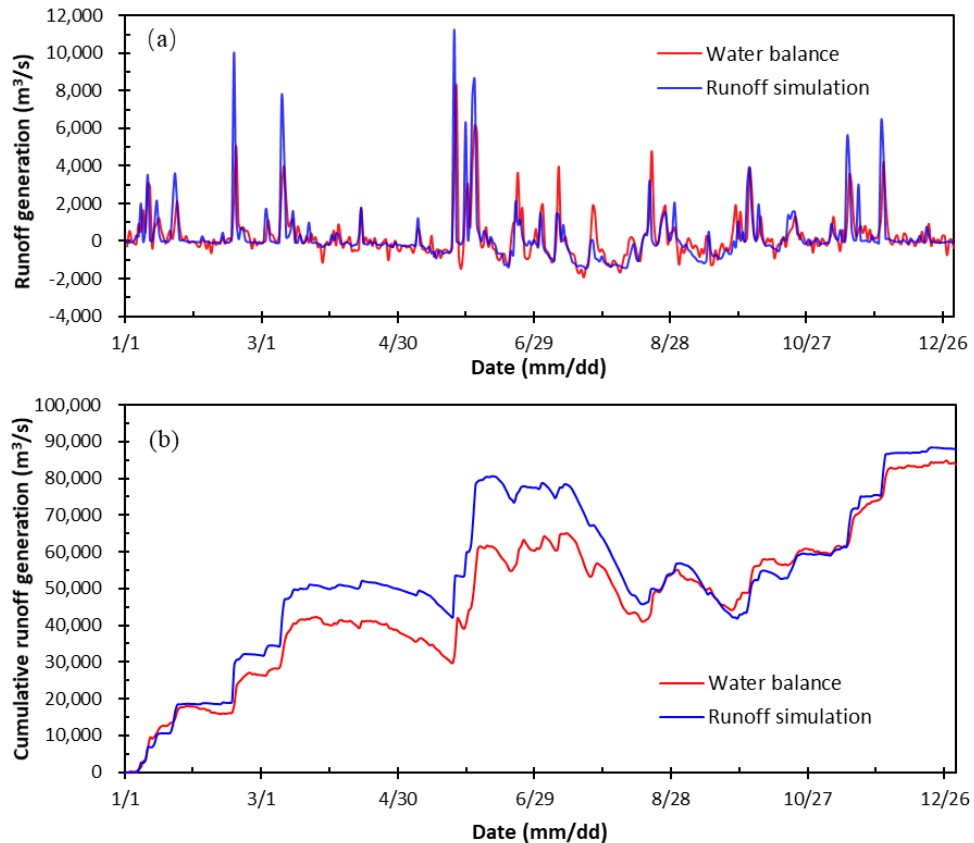

**Figure 8.** Time series (**a**) and cumulative (**b**) of runoff for the whole Taihu Basin in 2000.

3.1.2. Hydrodynamic Simulation

The observed water surface elevation of 2000 was used for the initial condition. The initial discharge for the river network was set to zero. To reflect the actual water flow movement, the observed flow data of 13 large gates along the Yangtze river and three entrances were applied for the boundary conditions. All these gates in Taihu Basin were operated according to practical rules. In general, these gates are used to control water levels in different areas. When the water level in the area is high and exceeding a certain value, these gates are used to drain off water; when the water level in the area is low, these gates are used to withdraw water from Yangtze River with a high tide level. For example, in non-flood season, when the water level in Taihu Lake is higher than 3.5 m, the water will be drained off through the gates around Taihu lake; when the water level of Taihu Lake is lower than 3.2 m, the gates will open and withdraw water from Yangtze River; in other cases, the sluice gates will be closed. During the flood season, when the water level of Taihu Lake exceeds the flood control water level at 8:00 in the morning, the flood control operation scheme shall be adopted.

For the whole Taihu Basin, 16 regional representative water stage gauging stations (Figure 4b) were used for the calibration. The simulations of the time series for water surface elevation (WSE) agree well with the observed ones (Table 5 and Figure 9). Mostly, the root-mean-square errors (RMSEs) are lower than 0.1 m, and the coefficients of determination ($R^2$) were larger than 0.8. For parts of Zhejiang Province including Jiaxing, Nanxun, Tongxiang, Wuzheng, and Xinshi, these grow double-cropping rice, but single-cropping rice is used for the simulation in the whole Taihu Basin. The needs of irrigation water in the paddy field irrigation period played an important role in these regions. That is a possible reason for the lower simulation accuracy. Also, four stream-gauging stations were chosen to calibrate the developed Taihu model (Figure 10). They followed the pattern of the observed data with small relative error, except for the Baishaoshan gauging station.

**Table 5.** Statistical error parameters of simulated and observed water surface elevation (m) and discharge (m³/s) in 2000 for Taihu Lake and typical sites.

| Water Stage Gauging Stations | MAE [a] (m) | RMSE [b] (m) | R² [c] |
|---|---|---|---|
| Taihu | 0.04 | 0.05 | 0.92 |
| Jintan | 0.14 | 0.18 | 0.82 |
| Liyang | 0.13 | 0.17 | 0.83 |
| Yixing | 0.05 | 0.07 | 0.85 |
| Fangqian | 0.09 | 0.11 | 0.83 |
| Wuxi | 0.05 | 0.08 | 0.88 |
| Fengqiao | 0.06 | 0.08 | 0.81 |
| Xiangcheng | 0.03 | 0.04 | 0.86 |
| Kunshan | 0.02 | 0.03 | 0.90 |
| Chenmu | 0.02 | 0.02 | 0.89 |
| Wangjiangjing | 0.01 | 0.02 | 0.93 |
| Jiaxing | 0.10 | 0.12 | 0.65 |
| Nanxun | 0.07 | 0.09 | 0.81 |
| Tongxiang | 0.13 | 0.16 | 0.65 |
| Wuzhen | 0.11 | 0.14 | 0.76 |
| Xinshi | 0.09 | 0.11 | 0.76 |
| **Stream Gauging Stations** | **MAE (m³/s)** | **RMSE (m³/s)** | **R²** |
| Luoshe | 11.18 | 13.80 | 0.71 |
| Baishaoshan | 7.20 | 8.89 | 0.51 |
| Fengqiao | 2.38 | 2.94 | 0.84 |
| Pingwang | 34.47 | 42.56 | 0.74 |

Note: [a] The mean absolute error, [b] The root mean square error, [c] The coefficient of determination.

### 3.2. Model Validation

The representative rainfall data from 1998,1999, 2002, and 2003 were used for validating the developed Taihu model. In 1998, the precipitation was higher than in normal years, and the whole of Taihu Basin suffered a severe flood in 1999. The precipitation in 2002 and 2003 was low. The same as the calibration process, the basic data of 1998, 1999, 2002, and 2003 included rainfall, evaporation, water surface elevation, discharge, tide level, water resource consumption, and discharge, and were all processed and imported to the DFBMS.

In general, the simulation results mimic the observed data to a large extent in these four years (Table 6). The best simulation of WSE in Taihu Basin was in 1999, then 2002 and 2003 came second, and the worst was in 1998. Take simulation in 1999 as an example, the simulated WSE was mostly higher than the observed ones in different locations (Figure 11). This is due to the extraordinary flooding in 1999, which resulted in many breakdowns in the districts of storing floodwater, low-lying land, and polder without collected data. Also, the discharge simulation was much closer to the observed data in 1999 compared with the results in other years (Table 6 and Figure 12).

Overall, the simulations of WSE and discharge for the representative stations were close to the observed data in these four years. Except for certain individual stations (such as Fengqiao, Tongxiang, and Wuzhen), the water surface elevation and discharge were in good agreement with the survey results. There were lots of factors affecting computational accuracy, such as precipitation, evaporation, underlying surface basic data, river data, date of operation, parameters, and so on. Furthermore, a lack of sufficient measured data led to errors in the parameter calibrations. In general, the modeling accuracy for the whole Taihu Basin was reasonable. The DFBMS reflected the characteristics of runoff generation and water movement in Taihu Basin.

### 3.3. Operation Rule Analysis

Water pollution problems in Taihu Basin become more and more serious due to the fast urbanization process. In 2001, the water conservancy department of China proposed a project of water diversion from Yangtze River to Taihu Lake to improve the water quality

in Taihu Basin, especially for Taihu Lake. The water flow in Taihu Basin was very complex under combined operations. The water environment mainly improved through the dispatch and operation of the hydraulic engineering structure of gate, dam, hydropower, and so on. In the simulations of 1998, 1999, 2000, 2002, and 2003, the numerical model was proven applicable in actual practice. Based on the calibration and validation, it can be used to simulate the water division from the Yangtze River to Taihu Basin. Hence, the DFBMS used the 2003 data to simulate and analyze the operation rule situations.

(a) Representative water stage gauging stations in Taihu lake

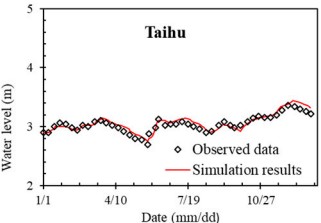

(b) Fifteen regional representative water stage gauging stations

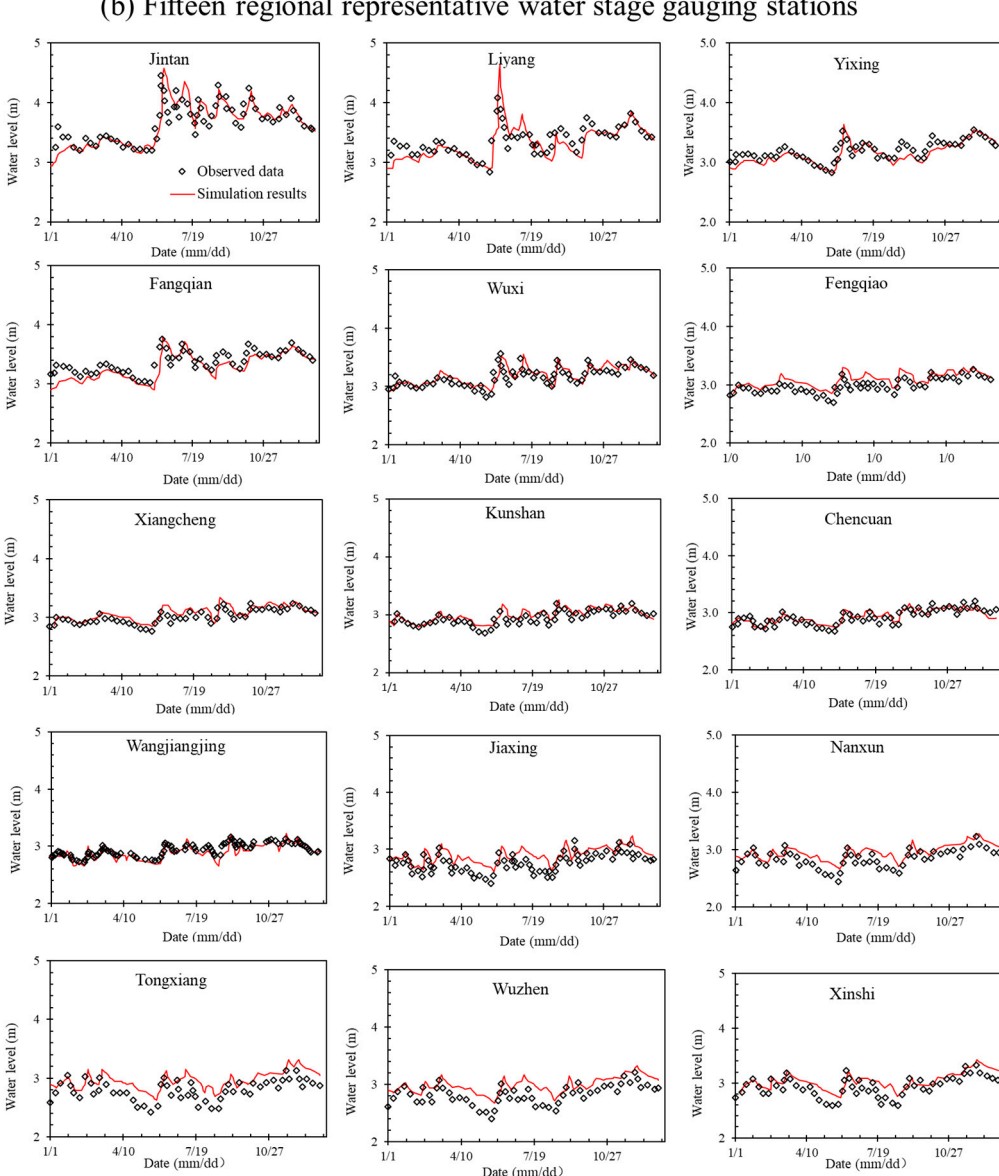

**Figure 9.** The comparison of simulated and observed water surface elevation in representative stations in 2000.

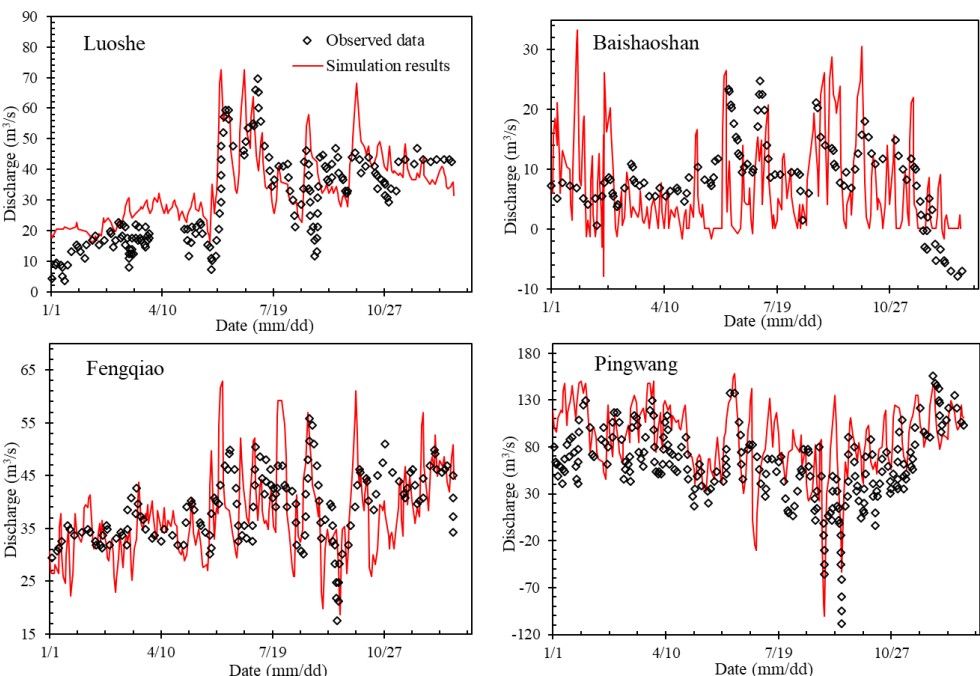

**Figure 10.** The comparison of simulated and observed discharge in Luoshe, Baishaoshan, Fengqiao, and Pingwang in 2000.

**Table 6.** Statistical error parameters of simulated and observed water surface elevation (m) and discharge (m³/s) in 1998, 1999, 2002, and 2003 for Taihu and typical sites.

| Gauging Stations | 1998 | | | 1999 | | | 2002 | | | 2003 | | |
|---|---|---|---|---|---|---|---|---|---|---|---|---|
| Water Stage | MAE [a] (m) | RMSE [b] (m) | R² [c] | MAE [a] (m) | RMSE [b] (m) | R² [c] | MAE [a] (m) | RMSE [b] (m) | R² [c] | MAE [a] (m) | RMSE [b] (m) | R² [c] |
| Taihu | 0.03 | 0.04 | 0.91 | 0.04 | 0.05 | 0.98 | 0.03 | 0.03 | 0.89 | 0.03 | 0.03 | 0.90 |
| Jintan | 0.12 | 0.16 | 0.76 | 0.13 | 0.16 | 0.90 | 0.16 | 0.19 | 0.74 | 0.08 | 0.11 | 0.85 |
| Liyang | 0.09 | 0.12 | 0.84 | 0.12 | 0.16 | 0.94 | 0.11 | 0.14 | 0.87 | 0.12 | 0.16 | 0.93 |
| Yixing | 0.09 | 0.11 | 0.91 | 0.07 | 0.09 | 0.93 | 0.06 | 0.07 | 0.95 | 0.05 | 0.06 | 0.93 |
| Fangqian | 0.12 | 0.15 | 0.72 | 0.08 | 0.10 | 0.91 | 0.10 | 0.12 | 0.79 | 0.07 | 0.09 | 0.91 |
| Wuxi | 0.07 | 0.09 | 0.73 | 0.05 | 0.07 | 0.93 | 0.06 | 0.07 | 0.85 | 0.09 | 0.13 | 0.74 |
| Fengqiao | 0.06 | 0.07 | 0.77 | 0.03 | 0.04 | 0.93 | 0.05 | 0.06 | 0.86 | 0.03 | 0.04 | 0.84 |
| Xiangcheng | 0.05 | 0.06 | 0.73 | 0.05 | 0.07 | 0.86 | 0.06 | 0.07 | 0.75 | 0.08 | 0.10 | 0.78 |
| Kunshan | 0.05 | 0.06 | 0.70 | 0.07 | 0.09 | 0.87 | 0.05 | 0.06 | 0.74 | 0.02 | 0.03 | 0.85 |
| Chenmu | 0.05 | 0.07 | 0.80 | 0.04 | 0.05 | 0.88 | 0.05 | 0.06 | 0.78 | 0.02 | 0.02 | 0.89 |
| Wangjiangjing | 0.11 | 0.13 | 0.75 | 0.09 | 0.12 | 0.83 | - | - | - | - | - | - |
| Jiaxing | 0.11 | 0.13 | 0.71 | 0.10 | 0.13 | 0.81 | - | - | - | - | - | - |
| Nanxun | 0.08 | 0.11 | 0.95 | 0.09 | 0.12 | 0.86 | - | - | - | - | - | - |
| Tongxiang | 0.11 | 0.14 | 0.78 | 0.13 | 0.17 | 0.62 | - | - | - | - | - | - |
| Wuzhen | 0.12 | 0.14 | 0.84 | 0.11 | 0.15 | 0.82 | - | - | - | - | - | - |
| Xinshi | 0.10 | 0.12 | 0.91 | 0.13 | 0.17 | 0.84 | - | - | - | - | - | - |
| Discharge | MAE (m³/s) | RMSE (m³/s) | R² | MAE (m³/s) | RMSE (m³/s) | R² | MAE (m³/s) | RMSE (m³/s) | R² | MAE (m³/s) | RMSE (m³/s) | R² |
| Luoshe | 7.59 | 9.49 | 0.79 | 6.92 | 8.99 | 0.85 | - | - | - | - | - | - |
| Baishaoshan | 21.80 | 27.25 | 0.68 | 17.43 | 22.64 | 0.63 | 16.70 | 20.11 | 0.59 | 6.76 | 9.02 | 0.71 |
| Fengqiao | 16.51 | 20.64 | 0.93 | 15.33 | 19.91 | 0.92 | - | - | - | - | - | - |
| Pingwang | 47.16 | 58.94 | 0.86 | 26.97 | 35.03 | 0.94 | 42.07 | 50.69 | 0.83 | 46.33 | 61.77 | 0.69 |

Note: [a] The mean absolute error, [b] The root mean square error, [c] The coefficient of determination.

For the diversion scenario, i.e., actual practice in 2003, Wangtinglijiao gate was open to divert water from Yangtze River at 278 m³/s from 08:00 on 8 August (Figure 13a). The location of Wangtinglijiao gate is shown in Figure 14. For the case of a no-diversion project, the Wangtinglijiao gate was kept closed. For the other gates in Taihu Basin, the operational states were set as the actual situation. The simulation period was from 1 April to 30 September. The water surface elevation at Taihu Lake under diversion and no-diversion scenarios is shown in Figure 13b. Obviously, the water surface elevation of Taihu Lake is higher when there is no diversion from the Yangtze River. The purpose of water diversion was to improve water quality, which includes variations in ammonia nitrogen, the five-day biochemical oxygen demand, phosphorus, and so on. Due to the length limit, water quality was not included and only the water quantity model is discussed in this study. The dye

distribution of the waterbody (in red) shows the diversion from Yangtze River at 8:00 on 15 August (Figure 14), which was the seventh day after diversion. During the diversion, the water also expanded to other river networks, which proves the complex water movement in the plain area. In general, the DFBMS gives general information on the water movement from Yangtze River to Taihu Lake, and the discharge from Taihu Lake to other regions.

## (a) Representative water stage gauging stations in Taihu lake

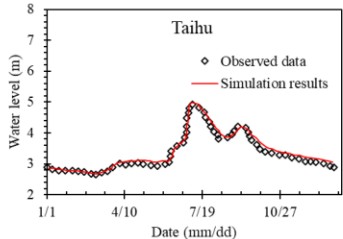

## (b) Fifteen regional representative water stage gauging stations

**Figure 11.** The comparison of simulated and observed water surface elevation in representative stations in 1999.

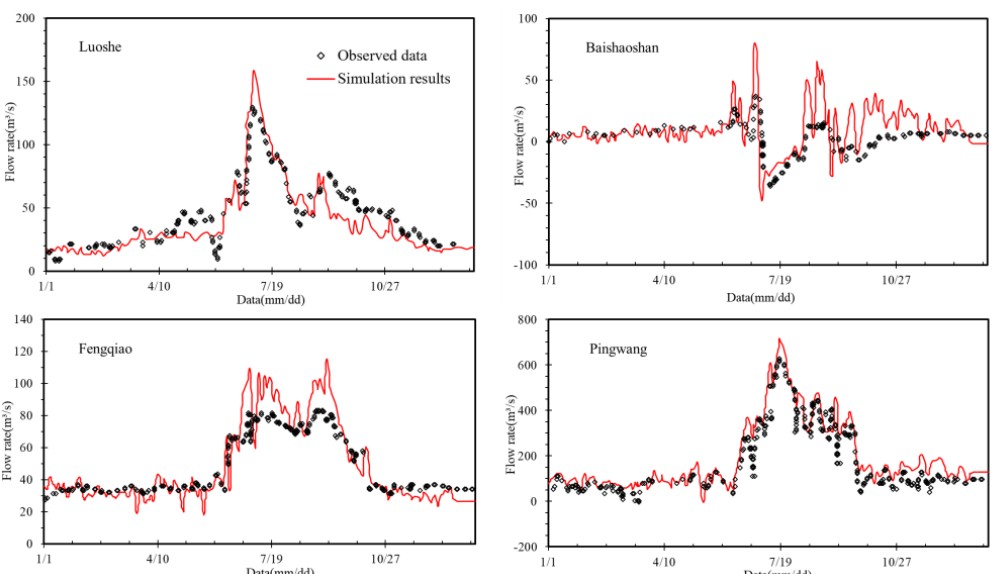

**Figure 12.** The comparison of simulated and observed discharge in Luoshe, Baishaoshan, Fengqiao, and Pingwang in 1999.

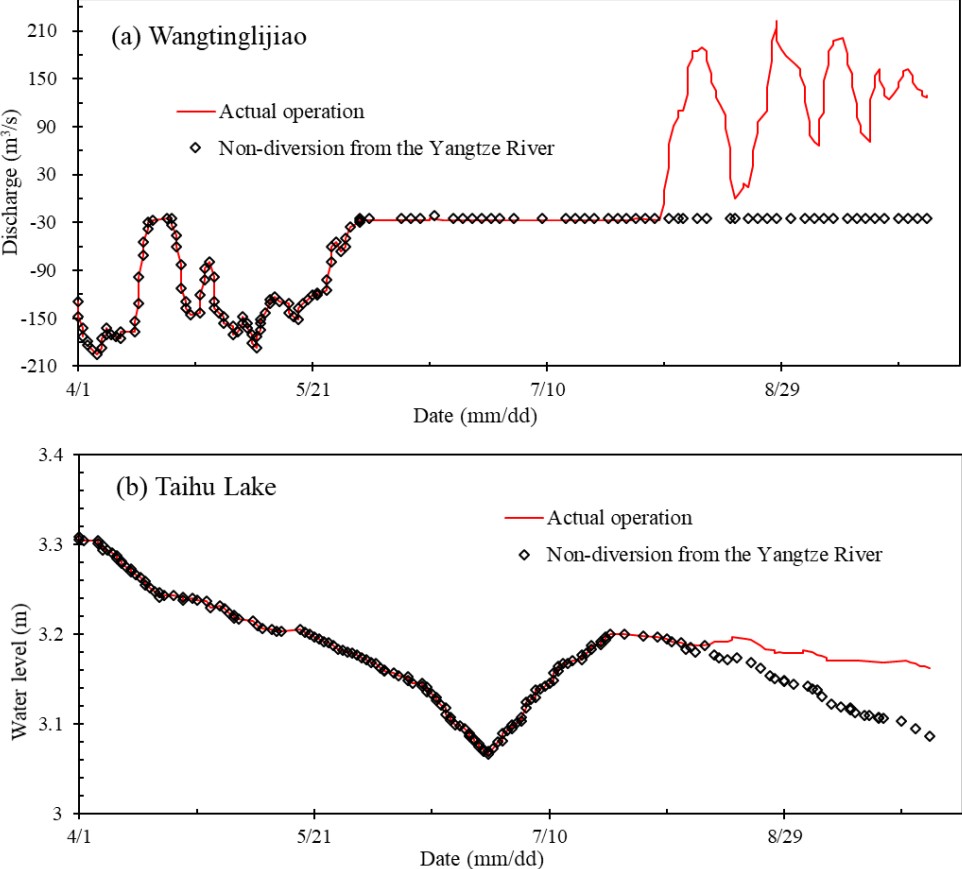

**Figure 13.** (**a**) The discharge at Wangtinglijiao and (**b**) the water surface elevation in Taihu Lake under conditions of diversion and non-diversion from the Yangtze River.

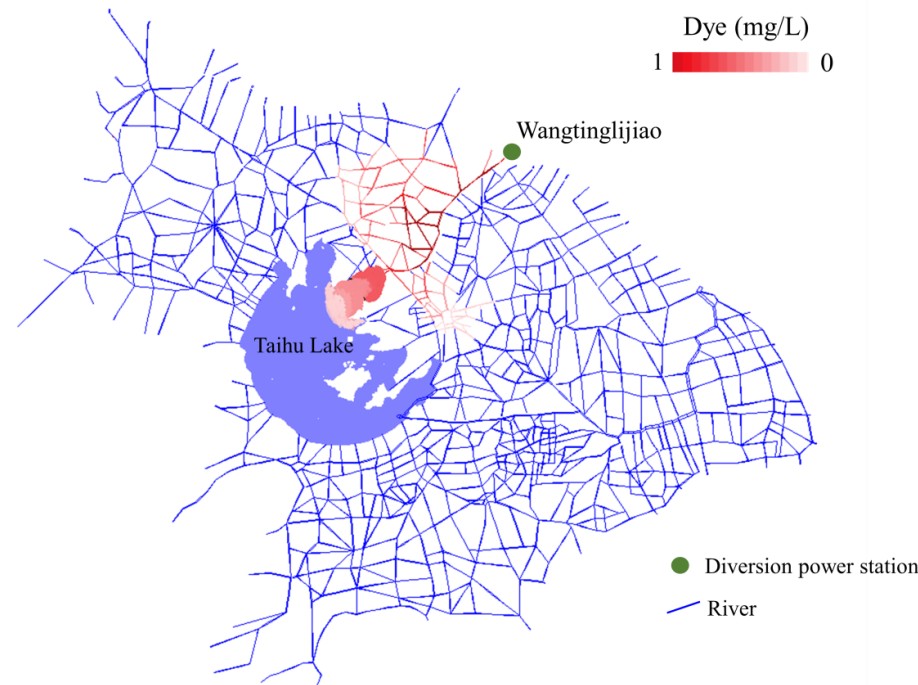

**Figure 14.** The distribution of water (in red) from Yangtze River under diversion through Wangtinglijiao at 8:00 on 15 August.

## 4. Summary and Conclusions

The DFBMS has three parts: a database management system, a professional model system, and an integrated geographical information system. Our four series papers mainly focus on the professional modeling system, the double-object sharing structure, and applications in different watersheds. DFBMS is the visual decision-making modeling system that can be used to simulate changes in the water cycle. The theoretical method of DFBMS is feasible, as has been proved through the application in the Three Gorges area (the 2nd series paper), Huai River Plain (the 3rd series paper), and Taihu Basin (the 4th series paper or this paper).

In the first paper, HFU was proposed and defined as a geographic area that has the same mechanism of runoff generation and confluence. HFU can be classified by various runoff types, confluence types, or mixed runoff and confluence. A study area can be divided into 11 different kinds of HFUs. A distributed-frame professional modeling system (DF-PMS) adopts different but mostly suitable submodules to simulate each HFU. The first paper also proposed a double-object sharing structure (DOSS), which is another foundation of the DFBMS. The sharing uniform data structure was used to integrate the hydrological model and GIS.

DF-PMS contains a distributed-frame hydrologic modeling system (DF-HMS) and a distributed-framework river modeling system (DF-RMS). In the second paper, the hydrologic HFUs of runoff generation and confluence that form DF-HMS were introduced. For the third paper, the numerical procedures that the DF-RMS uses to simulate the hydraulic processes of flow movement in river networks and lakes were discussed in detail. DF-HMS and DF-RMS were both applied in the specified study areas to test the model's reliability.

This last paper built a hydrologic and hydrodynamic model for the whole Taihu Basin, which is a lowland plain area with lots of polder areas. The simulation for Taihu Basin was quite complex and categorized into hilly sub-watersheds, hilly rivers, plain overland flows, plain rivers, lakes, reservoirs, and hydraulic engineering structures. The calibration and validation results proved that the DFBMS reflected the runoff generation and water movement accurately for Taihu Basin. This study provided a guide for the sustainable use of water resources and socio-economic development in the whole basin. For applications

of DFBMS in different study areas, the key step is to generalize the areas into many HFUs based on the different underlying surfaces. The hydrological processes and hydrodynamics can be coupled and simulated through water exchange in each HFU. Overall, DFBMS is a useful modeling system that can be applied to improve the response and handling abilities for real-time flood forecasting and other unexpected water disasters.

**Author Contributions:** The work was conducted by G.C., C.W., X.F., X.L., P.Z., and W.H.; this paper was written by G.C., C.W., and X.F. reviewed and improved the manuscript with comments; the data compilation and statistical analyses were completed by all authors. All authors have read and agreed to the published version of the manuscript.

**Funding:** This research has been financially supported by the National Key Research and Development Program of China (2018YFC1508200), Project 41901020 supported by NSFC, and the Fundamental Research Funds for the Central Universities (B200202030), and Hydraulic Science and Technology Program of Jiangsu Province (2020003).

**Institutional Review Board Statement:** Not applicable

**Informed Consent Statement:** Not applicable.

**Data Availability Statement:** Not applicable

**Conflicts of Interest:** The authors declare no conflict of interest.

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
