# Peer review of "Distributed-Framework Basin Modeling System: IV. Application in Taihu Basin"

_water, doi:10.3390/w13050611_

Round 1

Reviewer 1 Report

Thanks for responding to my comments. Good work.

Reviewer 2 Report

I truly believe that the 4 papers could be synthesized into just one single paper, but I’ll leave that decision up to the Editor. Having said this, the authors have addressed each of my concerns and they are applauded for improving the papers. I appreciate the contribution of the authors in reporting their hydrologic modeling framework and I encourage them to work in future implementations with public repositories with codes and tutorials written in English for easier access to the international community.

Reviewer 3 Report

The authors of the manuscript “Distributed-Framework Basin Modeling System: Application 2 in Taihu Basin (Ⅳ)” revised their manuscript according to the reviewers’ comments. The new output is an ameliorated version of the first manuscript and in my point of view the final output is very good scientific paper. Congratulation to the authors.

This manuscript is a resubmission of an earlier submission. The following is a list of the peer review reports and author responses from that submission.

Round 1

Reviewer 1 Report

General comments

As a continuation from the 2nd manuscript, and 3rd manuscript, the authors implemented their proposed modelling systems in a complex case study area, namely Taihu Basin. The case study area is well presented, as well as the proposed Hydrological Feature Units (HFUs) in Taihu Basin. However there are issues in the calibration and validation sections that need to be improved, in order to validate the accuracy of the modelling procedure. Based on my comments, I propose the major revision of the manuscript.

Specific comments

Line 111. Digital map instead of “electronic” map

Table 1. The authors should use smaller font size of the letters, because some figures are not properly presented.

Lines 143-144. “After we obtained the underlying surface information for each grid, the runoff on each surface was calculated based on the rainfall‒runoff mechanism introduced in the second paper in the series.”. Unfortunately, the description given in the second paper in the series is not adequate. You have to ameliorate the 2nd manuscript particularly in the sections that you speak about the rainfall-runoff simulation. On the other hand table 4 of the current manuscript gives more detailed information about the processes.  

Lines 151-154. “The application of the distributed-frame hydrological and hydraulic model in Taihu Basin includes different corresponding HFUs. Figure 4 shows the logical architecture diagram of the distributed-frame professional modeling system in the developed Taihu Basin model.” I suppose that in Figure 4 there are designated (highlighted with cyan frame) the types of HFUs. I do understand the runoff generation type, but I cannot understand what “rainfall and runoff” and “river-network, generalization ”stands for. These two elements have not been introduced in the previous manuscripts. Please provide more information.

Lines 155-156. “The runoff generation and confluence were simulated based on the rainfall and underlying surface data”. In continuation to my previous comment, in which way the rainfall runoff was simulated? Didn’t the authors use the models proposed in their respective HFUs?

Table 4. In the generalization details column of Table 4, the authors use the term subcatchments. What is the meaning then of introducing the hydrologic computing units (HCUs)? The subcatchments are not considered HCUs?

Additionally, in Table 4, the authors indicate the simulation of Taihu Lake area was conducted with a Quasi three-dimensional model from the 3rd paper. However, in the 3rd paper the authors present a two-Dimensional Flow Simulation in Lakes, in particular for the simulation of the flow movement. Is that a different model? If not, the authors should use the same terminology in all the submitted manuscripts, since they are related and thus the terminology should be constant.

Line 208. “The distributions of generalized polder areas in Taihu Basin are shown in Figure 6b.” Which is the Figure 6b. Please correct that.

Figure 7. Please remove the blue colour representing the sea.

Lines 235-239. “According to a survey report in 2000, there is 4.1 billion m3 water from 223 water supply companies and 11.8 billion m3 water from 118 self-contained water sources in the basin. As to the specific geographic location, there is 3.78 billion m3 water from 133 water supply companies and 11.28 billion m3 water from 102 self-contained water sources (Figure 8a).” The meaning is not clear. What do you mean by “there is 4.1 billion m3 water”. Water withdrawals? What is the difference between the first and second aforementioned sentences? Please rephrase.

Line 261. “3.1.1. Runoff calculation for Taihu Basin.” Although the methodology for infiltration is presented in the 3rd manuscript, information about the way that infiltration losses were computed in the case study area should be provided. Moreover, are the losses of the open channel structures for irrigation taken into account? If yes, in which way?

Additionally and regarding the results, statistical figures, e.g. coefficient of determination (R2) and the root mean square error (RMSE), etc, should be provided for the runoff simulation in order to see the correlation outputs.

Line 270. “The objective was to calculate the change of waterbody storage for the plain area”. What the authors mean with the term waterbody storage? The hydrosystem consists of rivers, lakes, irrigation and drainage channels etc, thus the term waterbody is not clear.

Lines 273-274. “…representative water stage gauging stations are chosen to calculate the daily mean water surface..”. The mean water surface of what? Of the lakes, of the rivers? By which way the water surface of the lake is combined with the mean water surface of a river stream?

276-277. “…where positive and negative means a storage increase to decrease, respectively”. What is the storage decrease? What is the physical meaning of this? This is also expressed in the Figure 9a, where there are negative values. What the authors mean with the term runoff generation and in which way the model gives negative values? What is the physical interpretation of negative values in the simulation of the runoff?

Figure 9. What the authors mean with the term runoff generation in the legend of the figure?

Line 291. “3.1.2. Hydrodynamic Simulation”. In these section the authors present their simulation outputs for the year 2000. However, the outputs presented in Table 5 are minimal. The authors compare just a single value for the whole year.  Statistical figures, e.g. coefficient of determination (R2) and the root mean square error (RMSE) etc (goodness of fit tests) , should be provided for the hydronynamic simulation in order to see the correlation outputs.

Line 295. “All these gates in Taihu Basin were operated according to practical rules.” Which are these practical rules and in which way the were integrated within the model?

Lines 307-308. “Also, four-stream gauging stations were chosen to calibrate the developed Taihu model (Figure 11). They all followed the pattern of the observed data, with small relative error.” Please provide the outputs of goodness of fit tests in order to support your argument.

Line 378. “3.2. Model Validation”. In these section the authors should provide diagrams as those of Figure 10b. Moreover, goodness of fit tests are required since the outputs in Table 6 demonstrate just a single value in the whole year. What about the models behavior for the whole year? That’s way the diagrams and goodness of fit test are required.

Line 365-366. “Obviously, the water surface elevation of Taihu Lake is higher when there is no diversion from the Yangtze River”. I cannot understand this argument and the simulation outputs. Since large water volumes are diverted from the Yangtze River to Taihu Lake the water level within the lake should be increased and not lower than the initial state, i.e. without the diversion.

Reviewer 2 Report

This series of 4 papers presents the description of a distributed basin modeling system composed of several components. The strength of the model laid on the flexibility and number of processes that can be integrated and modeled across the hydrologic and hydraulic components. In general, the papers are well-written, and the methods included in the hydrologic and hydraulic components are well presented. Having said this, I do not consider proper to present this work as a series of 4 papers, the overall structure looks more suitable for a dissertation document or a chapter book, but the presented format does not fit with the overall goal of a scientific paper, in which should maximize the synthetizes of methods, results, and discussions without losing accuracy and transparency, which reminds me the popular said: “I didn't have time to write a paper, so I wrote a book.” I encourage the authors to reconsider to condense the work into one single paper in order to show their wonderful work. Below, I’m describing my major and minor comments for all 4 papers.

[Major Comments]

  1. The 4 papers should be presented as a single paper. “Paper 1” can be easily synthesized as the introduction section, “Paper 2” and “Paper 3” is the section method, and “Paper 4” would be the Case Study. I noted through the papers several redundancies that can be avoided in order to achieve the best synthetizes in the work. For instance, “Paper 2” and “Paper 3” show a Case study; however, that should be the main purpose of “Paper 4”. There are several sections in “Paper 2” and “Paper 3” that can be moved to a Supplemental Material section (See Minor Comments).
  2. As I mentioned before, the strength of the study is found in the hydrologic and hydraulic components. However, “Paper 4” decreases the overall impact of the presented model. The authors were limited to show that the proposed model was able to replicate the discharge, water depth in some gauges just for a short period of time (calibration 1 year, validation 1 year). In general, there is not an analysis of the spatial distribution of the model performance, and there is no understanding of how the different model components, either hydrologic or hydraulic, is improving the representation of the hydrologic processes. This paper shows a new hydrologic model framework, therefore, should be expected to find an extended analysis of the different capabilities of the models showing the improvement of the model with and without different components
  3. The authors did not provide the source code or repository of the model. This is essential for future implementations of the model in the hydrologic community. In the case that the model is not available to the public, the authors must provide further details about the configuration in computational times used to run the simulations.

Minor Comments on: “Distributed-Framework Basin Modeling System: Overview and Model Coupling (I)”

[Line 25] What advantages? This statement is ambiguous

[Line 33] What does FH69 stand for?

[Line 106] Be careful using the argument of “Temporal GIS”, this is a matter of perspective, somebody could argue that including time-series to represent rainfall fields is sufficient to be in the realm of “Temporal GIS”. Besides, geological models do not seem an appropriate example to show the inconvenience of temporal representations in the current hydrological models, note that the geological processes evolve in a dramatic larger time scale in comparison to hydrologic processes explored in most of the hydrologic models.

[Figure 5] The captions need to be improved

Minor Comments on: “Distributed-Framework Basin Modeling System: Hydrologic Modeling System (II)”

[Line 20] Only two HFUs? what about the other 9 HFUs? Is there any model documentation?

[Line 122] What do SFD and MFD stand for?

[Section 2.1 2.1.1, 2.1.2] This section could be omitted or summarized in one or two paragraphs. Most of this content may be considered as general knowledge in the hydrologic community (e.g. estimation of flow direction D8), so there is no need to be so explicit in its development. Another option is included as Supplemental Material.

[Equation 2] What water depth and Chezy coefficient are used in Eq 2? Are they varying over space? Or is it just the DEM elevation used as the water depth in this case? Or are assumed constant across all the DEM processing?

[Lines 261-268] The runoff generation process and the overland flow must be explained in this section! The authors are just limited to provide some references; this is one of the key elements in the description of any hydrologic model.

[Lines 270-278] Be specific with the hydrograph method. This statement is vague, what equations and approximations are the authors using for the hydrograph routing method?

[Section 2.2.3] So does the plain overland runoff generation considers land use, but the Hilly- subwatersheds do not? Section 2.2.3 is nicely documented, however, section 2.2.1 is poorly described.

[Line 320] What specific parameter range? Be specific.

[Line 323] How necessary is to include this complexity in modeling runoff on the paddy fields? Have the authors provided any evidence of the adequacy in including this process? This should be explored in high detail in “Paper 4”

[Line 361] Provide ranges for Hp, Hu, and Hd

[Section 2.3] What about the modeling in woodland land use?

[Section 2.2.4] Include a description of the overland runoff method used. Again, what about the woodland surface?

[Section 2.3] This is not necessary if it is mentioned in “Paper 3”

[3 Study Case] This should be part of “Paper 4”

[Section 3.2] The evolution of the model performance needs to be improved. Please consider using the Nash–Sutcliffe model efficiency coefficient since has been used as a standard in the hydrologic community

[Section 3.2] Be specific in how the calibration was performed. What method? And what parameters were calibrated in this case study?

[Section 3.2] What was the computational time?

Minor Comments on: “Distributed-Framework Basin Modeling System: Hydraulic Modeling System (III)”

[Section 2] Large part of this paper could be included as an Appendix or Supplementals Information

[General] The equation numbering is incorrect, please verify.

[Lines 42-59] If there is no further discussion about these aspects through the paper, then this section should be removed.

[Section 2.4] This should be part of the “Paper 4”

[Line 443-Line 444] Rewrite “1982 cases...” for “case for the year 1982”; same for 1991.

[Figure 10] It is a better option to use a color bar to show the velocity field

Minor Comments on: “Distributed-Framework Basin Modeling System: Application in Taihu Basin (IV)”

[Figure 11] Are there only 4 streamflow gauges? Show statistics RMSE, Nash–Sutcliffe model efficiency

[Section 3.2] Why is the validation period and calibration period so short? I assume that there should be longer streamflow records within the basin, however, the authors only used one year for calibration and one for validation which obscures the true overall model performance that could be achieved with larger hydrologic records.

[General] Show the drainage area associated with each streamflow station

Reviewer 3 Report

This fourth paper is a case study of DFBMS in Taihu Basin. The authors did a good job in presenting the model calibration and validation results. The statistical analysis also makes sense. Overall I enjoyed reading it, but some of the figure presentations do need improvements. Please see my comments below:

1. Figure 1: this includes too much information to be read clearly. The color of low-elevation (light blue) is a bit mixed with rivers and lakes. I would suggest you to revise this figure to make it better. You may use Hybrid Google basemap or ESRI basemap. Mixing DEM colors with all the administrative information is not quite reader-friendly.
2. Figure 2 is a bit useless if you already show DEM in figure 1. Hilly and plain areas are already made clear in Figure 1 based on DEM.
3. Figure 7: What do all those colors mean? Why do you need different colors here? Can you make this a bit cleaner?
4. Figure 10: Does (b) have same legend as (a)? It's probably yes but can you make it clear in the figure or caption?
5. In your discussion section, can you also talk a bit more about how you would generalize this model for other areas?